# Chlamydia Pneumoniae Pericarditis Complicated with Guillain–Barré Syndrome

**DOI:** 10.3390/life15101532

**Published:** 2025-09-29

**Authors:** Vesna Ćosić, Đorđe Pojatić, Iva Bakalar, Nataša Moser, Karla Miškić, Blaženka Miškić

**Affiliations:** 1Faculty of Dental Medicine and Health Osijek, Josip Juraj Strossmayer University of Osijek, 31000 Osijek, Croatia; vesna.cosic@fdmz.hr (V.Ć.); natasa.moser@gmail.com (N.M.); miskickarla@gmail.com (K.M.); miskicblazenka@gmail.com (B.M.); 2Polyclinic Ćosić, 35000 Slavonski Brod, Croatia; 3General Hospital “Dr. Josip Benčević” Slavonski Brod, 35000 Slavonski Brod, Croatia; 4Department of Internal Medicine, General County Hospital Vinkovci, 32100 Vinkovci, Croatia; iva.bakalar@gmail.com; 5Dental Health Office “Dr. Karla Miškić”, 35000 Slavonski Brod, Croatia

**Keywords:** azithromycin, doxycycline, *Chlamydia pneumoniae*, Guillan–Barré syndrome, pericarditis

## Abstract

**Background:** Nearly 10% of all community-acquired pneumonias are caused by *Chlamydia pneumoniae*. This is a Gram-negative intracellular coccus that poses a significant challenge for routine diagnostics due to its poor growth in tissue culture and non-specific clinical presentations. **Case Report:** This study presents the case of a 61-year-old man whose initial disease manifestation included a non-specific upper respiratory tract infection and reactive pericarditis. A diagnostic work-up of the etiology of pericarditis, with stepwise exclusion of other causative agents, led to confirmation of a possible chronic, recurrent *C. pneumoniae* infection, with good clinical and laboratory responses to azithromycin across multiple hospitalizations. However, upon initiation of prolonged doxycycline therapy, the disease course was complicated by the development of Guillain–Barré syndrome. With appropriate treatment, the neurological deficit regressed, with near-complete resolution of the syndrome’s clinical picture. **Discussion and Conclusions:** The development of reactive pericarditis and the patient’s neurological symptoms stemmed from an immune response to bacterial antigens that resemble antigens of the central nervous system and the pericardium. Prolonged doxycycline therapy, together with symptomatic management of the neurological condition and concomitant pericarditis, represents good clinical practice and is one potential management approach for patients with similar presentations.

## 1. Introduction

The heart is enclosed by the pericardium, which consists of visceral and parietal layers. Among other functions, the pericardium coordinates the movement of the heart muscle in relation to other intrathoracic organs. Inflammation of the pericardium, or pericarditis, is a distinct cardiovascular occurrence.

According to current guidelines, it is characterized by the presence of at least two of the following four criteria: characteristic electrocardiographic changes, pericardial effusion, pleuritic chest pain, and/or a pericardial friction rub [1]. It may be of infectious, autoimmune, neoplastic, or idiopathic etiology, the latter being the most common. Pericardial inflammation is frequently accompanied by myocardial inflammation in the form of myopericarditis or perimyocarditis. Myocarditis arises from, among other mechanisms, autoimmune or cytolytic effects of infectious agents on cardiomyocytes and often results in fibrotic scarring of the tissue and the development of dilated cardiomyopathy [2]. Among the infectious etiologies of myopericarditis, intracellular bacteria merit particular attention, as they may present with a non-specific clinical picture and give rise to a broad spectrum of complications [3].

Chlamydia pneumoniae is a Gram-negative intracellular coccus identified in the second half of the twentieth century. Chlamydia pneumoniae belongs to the bacterial family Chlamydiaceae, along with Chlamydia trachomatis and Chlamydophila psittaci. It has two membranes, inner and outer, and is metabolically inactive outside of a host. The metabolically inactive form is called the elementary body. The reticulate body represents the active form, which is activated in host cells and divides via binary fission—for this purpose, it uses energy from host cells [4]. *Chlamydia* is transmitted via respiratory droplets and, after an incubation period of nearly three weeks, produces mild and atypical symptoms. It can infect endothelial cells of the pulmonary vasculature and be carried by neutrophils and macrophages to other parts of the cardiovascular system [5]. Through this mechanism, it can cause infections of the central nervous system and inflammatory disorders of the endocardium, myocardium, and pericardium, and it has been associated with atherosclerotic disease of the cardiovascular system.

Depending on the investigative methods used, some studies have confirmed the presence of *C. pneumoniae* in atherosclerotic plaques of the aorta and other large arteries in up to 30% of subjects [6]. The shared link between atherosclerosis and inflammatory states is reflected in the synergistic interplay between pro-inflammatory cytokines and inflammatory cells with other predisposing factors. Thus, dyslipidemia, obesity, and smoking act as accelerants that, in the setting of immune-cell activation by chronic infection, promote increased lipid uptake by macrophages, foam cell formation, and their deposition within the walls of the arterial vasculature [7]. The extent to which chronic infection promotes atherosclerosis is illustrated by the finding that, in individuals with no other risk factors, chronic chlamydial infection can increase the risk of an acute coronary event by up to fourfold [8].

The immunogenic potential of *Chlamydia pneumoniae* likely arises from similarity with antigens of the cardiovascular and central nervous systems, which trigger specific immune response mechanisms.

Although *C. pneumoniae* is presumed to frequently cause infections of the cardiovascular system, clinical evidence is largely limited to case reports [9]. A review of the literature indicates that the manifestation of chlamydial infection depends on the host’s immune response; in the context of a hyperreactive response, a case of acute heart failure presenting as cardiogenic shock and requiring extracorporeal circulation has been described [10].

*Chlamydia pneumoniae* is an intracellular coccus that is particularly difficult to cultivate in tissue culture; consequently, serological assays for the detection of antibodies to *C. pneumoniae* are regarded as the diagnostic gold standard [11]. IgA antibodies against *Chlamydia pneumoniae* indicate the presence of an active infection, and their short half-life of five to seven days allows a rapid assessment of infection intensity. The absence of IgA antibodies on serology, with concomitant presence of IgG antibodies against *C. pneumoniae*, is called seroconversion and indicates cessation of active infection [12].

Guillain–Barré syndrome (GBS) is an immune-mediated peripheral polyradiculoneuropathy characterized by the acute onset of symmetrical weakness (in the limb and axial musculature and in muscles innervated by the cranial nerves), sensory symptoms, absent tendon reflexes (areflexia), pain (shoulder girdle, back, buttocks, thighs), as well as respiratory failure and dysautonomia. The occurrence of asymmetric muscle weakness in GBS is uncommon and warrants particular caution to avoid misdiagnosis with other differentials (e.g., stroke, demyelinating myelopathy, acute-onset chronic inflammatory demyelinating polyneuropathy) [13]. Anti-ganglioside antibodies, congenital asymmetry in the development of the blood–brain barrier, as well as limb-use dominance, may play a role in the occurrence of asymmetric muscle weakness in the clinical picture of GBS [14]. Muscle weakness or paresthesia in GBS may begin in a single limb; however, most asymmetric symptoms at onset are transient and progress to symmetrical muscle weakness [15].

## 2. Case Report

The focus of our case is a 61-year-old man who underwent multiple hospitalizations during the autumn and winter of 2024 due to recurrent pericarditis.

His past medical history is notable for long-standing hypertension, hyperlipidemia, and atherosclerotic disease of the arterial vasculature at multiple sites. Prior to admission, non-obstructive atherosclerotic disease of the epicardial coronary arteries and obstructive atherosclerotic disease of the right internal carotid artery were diagnosed, and right internal carotid artery endarterectomy was performed. The patient also has chronic post-traumatic stress disorder (PTSD) consequent to trauma sustained during the Homeland War (Croatian War of Independence), for which he has been treated for many years with a combination of antidepressant therapies—escitalopram and mirtazapine.

### 2.1. First Hospitalization

On 6 September 2024, Mr. J.S. presented to the emergency department of General Hospital Vinkovci with fever (38.5 °C), dry cough, a maculopapular rash of the trunk, and a sense of general weakness within a two-day history of systemic infectious symptoms. Laboratory tests showed elevated C-reactive protein (158 mg/L), a left shift in the differential blood count (neutrophils, 91.5% by relative proportion; absolute count, 7.03 × 10^9^/L), signs of dehydration (creatinine 187 μmol/L, eGFR = 33 mL/min/1.73 m^2^), and hepatic enzyme elevations (AST 43 U/L, ALT 83 U/L, GGT 223 U/L) (Figure 1). A chest radiograph showed three nodular lesions of the pulmonary parenchyma measuring 3, 5, and 9 mm adjacent to the lateral thoracic wall, of indeterminate etiology. A diagnosis of acute tracheobronchitis was made, with a plan for diagnostic clarification of the nodular lesions. Abdominal ultrasonography excluded pathology of the hepatic parenchyma and of the other examined abdominal structures.

Parenteral therapy with amoxicillin/clavulanic acid and azithromycin was initiated immediately, alongside intravenous and oral rehydration, after which the patient became afebrile, with improvement in general condition and laboratory parameters. Computed tomography (CT) of the chest showed thin bilateral pleural effusions up to 10 mm and a pericardial effusion up to 7 mm, together with several non-specific subpleural nodules in both lungs measuring up to 10 mm in diameter. In view of convex ST-segment elevation in the lateral leads on electrocardiography, together with an echocardiographic finding of a hyperechoic pericardium and the presence of pericardial effusion, a diagnosis of pericarditis was established. Owing to his good general condition and regression of electrocardiographic and clinical signs of pericarditis, the patient was discharged on 11 September 2024 with a recommendation for outpatient follow-up and treatment with ibuprofen and colchicine.

### 2.2. Second Hospitalization

Readmission occurred on 28 October 2024. Two weeks earlier, the patient had, on his own initiative, discontinued colchicine and ibuprofen, after which he again became febrile and exhausted, with typical pericarditic chest pain that was less pronounced in the sitting position. Outpatient therapy with levofloxacin together with amoxicillin/clavulanic acid was commenced, but without improvement, and he re-presented for inpatient admission.

On admission, convex ST-segment elevation in the lateral leads was again documented, without reciprocal depression, together with a pericardial effusion and a hyperechoic pericardium. Laboratory tests again showed elevated inflammatory markers—131.2 mg/L of C-reactive protein—and a left shift in the differential blood count (neutrophils, 82.4% by relative proportion; absolute count, 8.7 × 10^9^/L); there were no signs of dehydration, myocardial injury (hsTnI 9 ng/L), or other pathology (Figure 1). As there was no improvement on the instituted antibiotic regimen, we reintroduced azithromycin together with colchicine and ibuprofen at therapeutic doses, after which the patient again became afebrile with a favorable regression of laboratory parameters.

Between the two hospitalizations, positron emission tomography (PET-CT) showed regression of the subpleural nodules in the pulmonary parenchyma, and tumor markers were within the reference range. PET-CT of the whole body was performed to exclude any malignant nature of subpleural nodules and to show the regressive or progressive dynamics of the nodules. The work-up identified an enlarged cervical lymph node for which fine-needle aspiration was planned; evaluation for possible tuberculosis and sarcoidosis was also scheduled. Given the presence of recurrent pericarditis, at the start of the second hospitalization, laboratory tests were conducted to investigate the immunological and microbiological etiology (Figure 2). The patient was discharged on 5 November 2024 with a plan for continued outpatient follow-up.

### 2.3. Third Hospitalization

Despite regular adherence to therapy, owing to an exacerbation of fever, recurrence of dry cough, and a generalized maculopapular rash, Mr. J.S. re-presented on 26 November for repeat cardiology assessment at General Hospital Vinkovci. The clinical picture was again dominated by pericarditic chest pain, with clear ST-segment elevation in the lateral leads and the presence of pericardial effusion. Laboratory tests again showed rising inflammatory markers (C-reactive protein = 52.9 mg/L) and a left shift in the differential blood count (neutrophils, 90.0% by relative proportion; absolute count, 9.82 × 10^9^/L), with other laboratory parameters within normal limits (Figure 1).

Guided by prior experience and the favorable response to azithromycin, the patient was administered azithromycin as parenteral monotherapy on the first day of hospitalization, leading to prompt defervescence. Over the following days, under azithromycin monotherapy, there was regression of the other symptoms as well, including the rash, dry cough, and pericarditic chest pain.

The previously requested laboratory, microbiological, and serological results are presented in Table 1. Fine-needle aspiration of the cervical lymph node showed features of a reactive inflammatory process, with no malignant cells identified.

Immediately prior to initiation of azithromycin therapy, four sets of blood cultures were obtained in accordance with endocarditis diagnostic guidelines; all remained sterile (no growth) [16]. Transesophageal echocardiography was performed and revealed no vegetations on the aortic or mitral valve leaflets.

Serology results were positive for *Chlamydia pneumoniae* in the form of positive IgA and IgG titers.

In view of the serological findings supporting chronic *Chlamydia pneumoniae* infection and the favorable clinical response to azithromycin, we elected to continue with oral doxycycline as the drug of choice, as recommended in the rare cases of chronic *C. pneumoniae* infection described to date.

The patient again became afebrile on the above treatment; however, owing to the onset of lower-limb muscle weakness, we deferred discharge. He refused the recommended neurology consultation and requested discharge on 3 December 2024, which he acknowledged by signing the designated form after the risks and rationale had been clearly explained.

### 2.4. Fourth Hospitalization, Neurology Department

On 8 December 2024, Mr. J.S. presented to the emergency department of General Hospital Vinkovci with worsening of previously noted lower-limb weakness. Since discharge from the Internal Medicine Department, his condition had continued to deteriorate.

On admission, neurological examination showed that he was alert, well oriented, and with normal speech; he understood and followed commands. The neck was supple, and he was negative for meningeal signs. Cranial-nerve assessment revealed reduced wrinkling of the right side of the forehead, incomplete closure of the right eye, a shallower right nasolabial fold, and lag of the right oral commissure on smiling—consistent with right-sided peripheral facial palsy (House–Brackmann grade III). He maintained both arms in an antigravity position, but gross muscle strength in both hands was reduced (4/5 on the Medical Research Council Scale (MRC scale)). He flexed both legs at the knees and, with effort, could lift each individually off the bed and briefly maintain it in the Mingazzini position, after which it sank back onto the surface. Plantar and dorsiflexion of both feet were preserved. Myotatic reflexes were absent in the lower limbs, with normal reflexes in the upper limbs, and there was no pathological plantar response. He reported no sensory disturbances, and there was no sensory-level deterioration. He could be brought to a standing position with the assistance of two people but was unstable on his feet; with effort, he walked with a wide-based gait, dragging his feet along the floor. Independent walking was not possible. No neurological deficit was found on coordination testing.

Urgent laboratory tests were performed, and the results were within reference ranges.

As part of the emergency work-up at General Hospital Vinkovci, an urgent computed tomography (CT) scan of the brain was performed, which showed no signs of demarcated acute ischemia, hemorrhage, suspicious expansions, or extra-axial collections.

On the second day of hospitalization, electroneuromyography (ENMG) was performed, supporting a diagnosis of acute polyradiculoneuritis—Guillain–Barré syndrome.

Subsequently, during hospitalization in the Neurology Department, magnetic resonance imaging (MRI) of the brain was performed, demonstrating chronic gliotic changes of vascular etiology, and MRI of the cervical spine showed absolute stenosis of the spinal canal from levels C3–4 to C6–7.

As part of the neurological work-up for acute polyradiculoneuritis, a lumbar puncture was performed, yielding clear, colorless cerebrospinal fluid (CSF) with albuminocytologic dissociation (protein 1796 mg/L) and no significant cellular content (leucocytes 2.0; erythrocytes 1.0). The Pandy test was positive (2+). Chloride and glucose levels were 127 mmol/L and 4.0 mmol/L, respectively. CSF cytology, microbiological, and serological analysis results were unremarkable (Table 2).

Among other laboratory findings, we noted a normal serum protein electrophoresis (total protein, 69.9 g/L; electrophoretic fractions: Alb, 46.4%; EF A1, 4.3%; EF A2, 9%; EF ß, 13%; EF GG, 27.3%). The levels of thyroid hormones, vitamins B12 and B9, and serum immunoglobulins (IgG, IgA, IgM) were within normal limits.

During hospitalization in the Neurology Department, there was a mild progression of motor weakness in the left leg. The patient reported numbness throughout the limb; he only minimally dragged it along the bed surface, was unable to lift it, and reported worsening pain in the lumbosacral spine. The remainder of the neurological examination was unchanged. Following consultation with an infectious diseases specialist and a cardiologist, it was concluded that, in this case, the benefits of intravenous immunoglobulin outweighed the potential risks; a total dose of 160 g was administered over five days. The therapy was well tolerated, without significant adverse effects, and an initially improving trend in the neurological deficit was documented (improved mobility of the left leg). Intensive physiotherapy was conducted during the hospital stay. The patient declined inpatient or spa rehabilitation.

On discharge, neurological examination showed residual right-sided peripheral facial palsy (House–Brackmann grade II). He maintained both arms in an antigravity position; he held the right leg in the Mingazzini position with mild oscillations, whereas he could only briefly lift the left leg, maintaining it for 2–3 s before it dropped back onto the bed. Plantar and dorsiflexion of both feet were preserved. The examination was dominated by reduced global muscle strength in the proximal muscle groups of the legs, with absent deep tendon (myotatic) reflexes in the lower limbs. At discharge, he was able to stand independently and walk with the aid of a walker.

It was recommended that he undergo further investigations, a neurosurgical evaluation, as well as follow-up electroneuromyography (ENMG) of the upper and lower extremities.

At subsequent outpatient neurology follow-ups, the patient showed satisfactory recovery to independent mobility, with no cranial nerve deficits and no motor weakness of the limbs; residual subjective symptoms were limited to mild numbness of the fingertips and toes. Electroneuromyography did not show any significant changes compared with the initial study. In monitoring his known atherosclerotic disease, follow-up color Doppler ultrasound of the carotid arteries showed an unchanged, previously known moderate stenosis of the left internal carotid artery (40–50%), while the right internal carotid artery was satisfactory, with no signs of postoperative restenosis.

## 3. Discussion

There are several important points that must be clarified in patients with chronic *C. pneumoniae* infection. Given the difficulty associated with confirming *C. pneumoniae* infection, it remains uncertain whether our patient was truly infected by this pathogen or whether the findings reflected a chronic infection. *C. pneumoniae* is an intracellular pathogen that is almost impossible to culture, and polymerase chain reaction (PCR) confirmation requires appropriate samples of infected tissue—something that is scarcely achievable in routine clinical practice. However, the simultaneous presence of IgA and IgG antibodies is considered a sufficient diagnostic criterion for chronic infection [12]. A second argument supporting this bacterium as the cause of the patient’s symptoms is the clinical course: on several occasions, there was no clinical or laboratory evidence of response to antibiotic therapy with agents such as amoxicillin/clavulanic acid, which is ineffective against this organism [17]. There was a near-immediate cessation of fever following intravenous azithromycin, and no flare of infection during a 60-day course of doxycycline, which is also a drug of choice for treating this infection. We decided to treat recurrent pericarditis with a combination of colchicine, ibuprofen, and the mentioned antibiotics—because pericarditis occurs in response to the infectious process—thus avoiding corticosteroids. The question remains whether the use of corticosteroids would lead to faster resolution of symptoms and prevention of neurological consequences. Treatment with corticosteroids can also lead to disease remission and more serious symptoms, as described in the guidelines for the treatment of pericardial diseases [1]. Our treatment strategy represents an innovative therapy with a good outcome and raises many new questions. A third, less obvious argument is the presence of atherosclerotic disease of the coronary and carotid arteries in a patient who was a non-smoker and had no other risk factors—unless chronic *C. pneumoniae* infection had been present for a prolonged period [18]. In similar reported cases with a fulminant onset culminating in shock, *C. pneumoniae* serology was negative, and infection was confirmed via PCR amplification of DNA extracted from tissue obtained via biopsy of the affected heart [19]. The indolent course in our patient supports the presence of a long-standing chronic infection that reactivated following impairment of the immune system by other factors. In addition, a fourth argument that the condition was caused by this pathogen is the extensive laboratory, microbiological, and imaging work-up, which excluded other infectious, immunological, inflammatory, and malignant processes that could result in similar symptoms. With reference to other published case reports, it may be inferred that molecular mimicry triggered a cross-reactive immune response against the patient’s pericardium and pleura, resulting in the clinical picture of pericarditis with reactive lymphadenopathy, subsequently leading—later in the disease course—to immunologically mediated Guillain–Barré syndrome [8]. Bachman et al. found that molecular mimicry can cause immune-mediated reactions in people infected with Chlamydia pneumoniae. Injection of part of the bacterial membrane whose amino acid sequence was homologous to that of β-myosin caused an autoimmune cytolytic reaction of lymphocytes against heart myocytes in experimental mice that were not infected with Chlamydia pneumoniae. Infiltration of CD8^+^ T lymphocytes led to an inflammatory reaction of the perivascular tissue and fibrosis of the affected part of the heart tissue [20]. Similar cardiac manifestations were observed in people infected with COVID-19, where the viral infection caused the myopathy of skeletal muscles and the development of myocarditis [21].

In addition to the distinct clinical recovery in response to the instituted treatment and the absence of systemic symptoms of infection, our results also demonstrated the absence of seroconversion. Similar observations have been reported in prior studies that used cultures from nasopharyngeal swab specimens, in which eradication of the pathogen was not achieved in all microbiological cultures [17]. However, a clear clinical response—in the form of cessation of symptoms, regression of symptoms, and resolution of disease manifestations in other organs—was considered a favorable response to treatment. The long-lasting outcomes of treatment with antibiotics such as tetracyclines and macrolides underscore the need for the development of new antibiotics with quicker action and better effect against obligate intracellular bacteria. These hypothetical molecules, such as macrolides and tetracyclines, should have high bioavailability following peroral ingestion. In addition, they should have both bactericidal and bacteriostatic effects on protein synthesis in prokaryotic cells because the elementary bodies of Chlamydia pneumoniae can survive, leading to disease relapse. The potential antibiotics should also have good pharmacokinetic characteristics against the membranes of the elementary bodies.

Guillain–Barré syndrome (GBS) is typically preceded by an infection or immune reaction that triggers an autoreactive immune response targeting the peripheral nerves and their spinal roots. GBS most often develops several days to weeks after a preceding respiratory or gastrointestinal infection [15]. It presents as a post-infectious ascending polyneuropathy with sensory symptoms and muscle weakness, which can progress to tetraparesis and dysautonomia [13]. In the patient presented here, *Chlamydia pneumoniae* infection was associated with the development of Guillain–Barré syndrome.

Atypical forms, such as asymmetric cranial nerve involvement, occur in some patients with GBS; this was the case in our patient. There was an infranuclear lesion of the seventh cranial nerve and tetraparesis with predominant involvement of the proximal musculature of the lower limbs, which was more pronounced in the left leg [22].

Although GBS is more commonly associated with infections caused by other pathogens, our clinical case demonstrates Guillain–Barré syndrome induced by *Chlamydia pneumoniae* infection—a presentation that remains rare in clinical practice [23].

In this case, due to further deterioration of the clinical picture consistent with acute polyradiculoneuritis, we opted for treatment with intravenous immunoglobulin (IVIg) at a dose adjusted to the patient’s body weight. Therapy lasted five days, and a good improvement in the neurological deficit was observed.

IVIg is a proven effective therapy for GBS. It is administered at a total dose of 2 g/kg body weight and is the preferred treatment in most facilities due to its availability and ease of administration [24].

Given that *Chlamydia pneumoniae* causes nearly 10% of all pneumonias in the general population and may act as a trigger for asthma, multiple sclerosis, GBS, and other chronic diseases, the true impact of this pathogen remains unclear and should be clarified using cross-sectional or cohort studies [4].

## 4. Conclusions

*Chlamydia pneumoniae*, which accounts for approximately 10% of all community-acquired pneumonias in the general population, can—among immunologically predisposed individuals—give rise to a range of reactive extrapulmonary manifestations, as illustrated by this case report. The data presented here indicate significant difficulties in the diagnosis of patients infected with Chlamydia pneumoniae. Collecting tissue samples for PCR is often difficult, and the only way to diagnose an infection is via serological methods, which are time-consuming, hence delaying treatment. Seemingly benign pathogens such as Chlamydia pneumoniae are an accelerating cause of cardiovascular and immunologically mediated diseases. The diagnostic and therapeutic approaches described here may help patients in similar clinical scenarios.

## Figures and Tables

**Figure 1 life-15-01532-f001:**
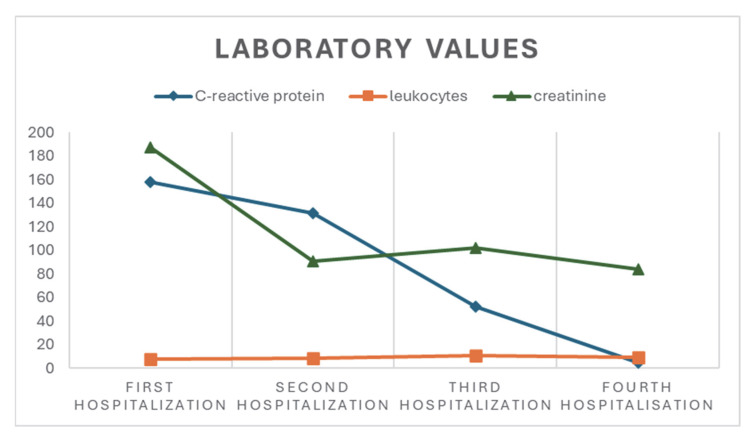
Basic laboratory values for the patient.

**Figure 2 life-15-01532-f002:**
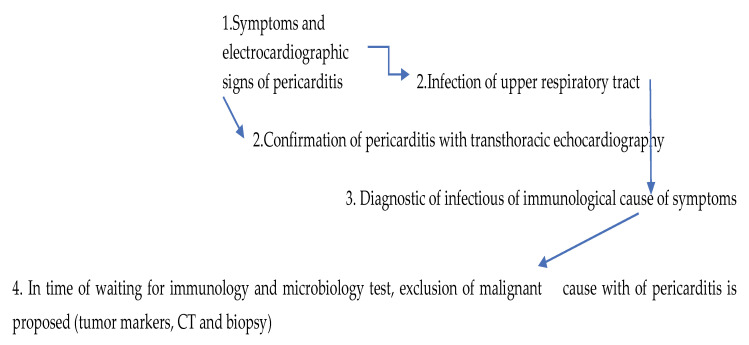
Diagnostic workflow.

**Table 1 life-15-01532-t001:** Serological and immunological work-up of the patient.

Laboratory, Microbiological, and Serological Tests	Results
Sputum for *Mycobacterium tuberc*. and QuantiFeron test	negative
Urine and blood cultures	sterile
Nasal swab for adenovirus antigen	negative
Throat swab (microbiology)	normal mucosal flora
Hepatitis B serology	anti-HBs positive; anti-HBc, HBaAg neg.
Hepatitis C serology	anti-HCV neg.
HIV serology	negative
*Borrelia burgdorferi* serology	IgG pos.; IgM negative
*Mycoplasma pneumoniae* serology	IgG pos.; IgM negative
Parvovirus B19 serology	IgG pos.; IgM negative
Cytomegalovirus serology	IgG pos.; IgM negative
Epstein–Barr virus serology	negative
***Chlamydia pneumoniae* serology**	**IgA (128) and IgG positive (>512)**
ANA, anti-dsDNA, ENA screening, anti-SS-A-/Ro, Anti-SS-B_7La, Anti-SM, Anti-SM/RNP, Anti-JO, Anti-Scl-70, Anti ribosomal antibodies, Anticentromere antibodies, Anti-U1-RNP, Anti-PM-Scl, ANTI-PCNA, Anti-histone, Anti-MPO (Panca), Anti-PR-3 (Canca), Anti-CCP antibodies	negative

**Table 2 life-15-01532-t002:** Neurological work-up of the patient.

Type of Laboratory Test	Measured Value and Meaning
Total CSF protein	1.57 g/L (above reference values)
CSF albumin	883 mg/L (above reference values)
Serum albumin	34.3 g/L (within reference values)
Q (CSF/serum)	25.8 (above reference values)
CSF IgG	302 mg/L (above reference values)
HSV serology, serum	IgM negative, IgG positive
HSV serology, CSF	IgG negative
*Borrelia burgdorferi* serology, CSF	IgM and IgG neg.
Anti-myelin-associated glycoprotein (anti-MAG)	negative
Anti-ganglioside antibodies (GM1, GT1a, GD1a, GD1b, GQ1b)	negative

## Data Availability

The data published in this study are available from the corresponding author.

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
