# Peer review of "Chlamydia Pneumoniae Pericarditis Complicated with Guillain–Barré Syndrome"

_life, 2025, doi:10.3390/life15101532_

Round 1

Reviewer 1 Report

Comments and Suggestions for Authors

Dear Authors,

Your Case Report entitled "Chlamydia pneumoniae pericarditis complicated with Guillain-Barré syndrome" has been carefully reviewed,

The Paper is well Written in English, and well presented. Tables and ideas are clear for readers.

As a general overview of the paper, I think that this paper should be revised by an English native speaker. Also some abbreviations must be explained for readers. Some figures related to the case should be added. In the Discussion it would be important to highlight on the idea of finding new antibacterial drug to fight this kind of bacterial infections.

Unfortunately, I do not see that this Case Report is suitable for publication in this journal, since it didn't represent a new case in the literature, since a similar case report entitled "Guillain-Barré syndrome after Chlamydia pneumoniae infection" was already published in February 1992 in the famous journal New England Journal of Medicine (NEJM).

Best Regards,

Author Response

Thank you very much for taking the time to review this manuscript. Please find the detailed responses in the attachment.

Reviewer 2 Report

Comments and Suggestions for Authors

Estimated Authors,

I've read with interest the present case report from Cosic et al. on a patient which developed a M pneumoniae infection with pericarditis and subsequent GBS. The case is substantially well described, and the central section of the paper can be only marginally improved - for instance, by expanding the range of Table 1 by also including laboratory data starting from Admission 1, and following ones.

Regarding introduction and later sections, the paper can be conversely improved by following the subsequent recommendations: (1) simplify the main text, particularly the introduction, by shortening it; Authors should also provide some more systematic data on M pneumoniae (e.g. M pneumoniae is a species of small gram-positive bacteria belonging to the family of Mycoplasmataceae and order Mycoplasmatales; M pneumoniae is among the smallest self-replicating organisms, with a reduced genome, limited metabolic activity etc...); (2) avoid some non-plain statements such as "enigmatic microrganisms" and similar ones; (3) in general the paper is quite long for a case report; please take this issue into account while reframing the paper.

Author Response

(The authors gave the same response as above.)

Reviewer 3 Report

Comments and Suggestions for Authors

Thank you for the opportunity to review this paper. The authors present a case of Chlamydia pneumonia recurrent infection complicated by pericarditis and GBS. This is a unique case highlighting the strong relationship between different infections and possible autoimmune sequela. Still, some issues should be addressed:

  1. While the intro is interesting, I think it should be more concise.
  2. Please explain why PETCT was performed after the first hospitalization?
  3. Did the ECG normalized after each treatment and then became pathologic again upon recurrence?
  4. If recurrent pericarditis was suspected in the second and third hospitalizations, why azithromycin was prescribed? Even if the patient had a good response to this treatment in the past, without knowing the source for his condition, this is a rather odd treatment for a patient with recurrent pericarditis.
  5. Discussion – the combination of pericarditis and GBS as a possible response to infection could be of autoimmune nature. This was shown by a case series in patients with COVID-19 having this type of combination of syndromes (DOI: 10.1007/s00296-022-05106-3). I recommend the authors to address this interesting aspect with the provided research and others.

Author Response

(The authors gave the same response as above.)

Reviewer 4 Report

Comments and Suggestions for Authors

The manuscript entitled Chlamydia pneumoniae pericarditis complicated with Guillain-Barré syndrome represents good clinical pratcies could be one potential management approach for patients with similar presentations.

Table 1 is not clear. Please make it a better representation.

Table 2. Neurological work-up of the patient—Needs to be represented in a better way

Why was the serum IgG level mentioned in the neurological work-up of the patient?

The conclusion should be more elaborate.

Author Response

(The authors gave the same response as above.)

Round 2

Reviewer 1 Report

Comments and Suggestions for Authors

Dear Authors,

The revised version of your Case Report entitled "Chlamydia pneumoniae pericarditis complicated with Guillain-Barré syndrome" has been carefully reviewed.

The paper lacks Figures for the case.

The paper is similar to the one I talked before about it "Guillain-Barré syndrome after Chlamydia pneumoniae infection"

Best Regards,

Author Response

Comments: "The revised version of your Case Report entitled "Chlamydia pneumoniae pericarditis complicated with Guillain-Barré syndrome" has been carefully reviewed.The paper lacks Figures for the case.The paper is similar to the one I talked before about it "Guillain-Barré syndrome after Chlamydia pneumoniae infection"."

Response: Due to your comments, we have added a new figure about the diagnostic workflow of our patients in case report article. We have also improved our English. 

Although you consider our article quite similar to one previously published in the "New England Journal of Medicine," we believe we have a unique treatment and diagnostic approach. 

Reviewer 3 Report

Comments and Suggestions for Authors

Thank you for the opportunity to review this paper once again. The authors have provided thoughtful and in-depth responses to all my comments and manuscript has significantly improved. I do not have further issues. Thank you and good luck!

Author Response

Comments: "Thank you for the opportunity to review this paper once again. The authors have provided thoughtful and in-depth responses to all my comments and manuscript has significantly improved. I do not have further issues. Thank you and good luck!".

Response:  Thank you very much. 

We uploaded a final version of the article, improved in English and corrected according to your previous demands.

Reviewer 4 Report

Comments and Suggestions for Authors

None

Author Response

Comments: None

Response: Thank you for your constructive criticism. We have uploaded a final version of the article, improved in English, according to your opinion that the English could and must be improved.